# The cost-effectiveness of penicillin allergy testing: Evidence and gaps from a systematic review

Ruben Ernesto Mujica-Mota[1,2]*, Miaoqing Yang[2], Natalie King[2], Shadia Ahmed[3], Neil Powell[4], Sue Pavitt[5], Bethany Shinkins[6], Jonathan A.T. Sandoe[7,8]

1 Health Economics Group, Department of Health and Community Sciences, University of Exeter, Exeter, United Kingdom, 2 Academic Unit of Health Economics, Leeds Institute of Health Sciences, University of Leeds, Leeds, United Kingdom, 3 Leeds Institute of Medical Research, School of Medicine, University of Leeds and Leeds Teaching Hospitals NHS Trust, Leeds, United Kingdom, 4 Pharmacy Department, Royal Cornwall Hospital, Truro, United Kingdom/ School of Biomedical Sciences, University of Plymouth, Plymouth, United Kingdom, 5 Dental Translational and Clinical Research Unit, University of Leeds, Leeds, United Kingdom, 6 Division of Health Sciences, University of Warwick, Coventry, United Kingdom, 7 Leeds Institute of Medical Research, University of Leeds, Leeds, United Kingdom, 8 Leeds Teaching Hospitals NHS Trust, Leeds, United Kingdom

* R.E.Mujica-Mota@exeter.ac.uk

## Abstract

### Introduction

Patients with a penicillin allergy label (PAL) use more and broader-spectrum antibiotics, have worse health outcomes and cost more to treat than patients without a PAL. A significant proportion of penicillin allergy labels are incorrect; here we review the published evidence on the costs, health-related quality of life, and cost-effectiveness of penicillin allergy testing.

### Methods

We conducted a systematic review of published economic evaluations of penicillin allergy testing in accordance with Cochrane guidelines. We searched Medline, Embase, Scopus, Web of Science, EconPapers (RePeC) and the International HTA Database (INAHTA) and included reports of full or partial economic evaluations of costs and/or health benefits of penicillin allergy testing strategies. The outcomes of interest were healthcare resource use, medical costs, and health-related quality of life for patients with a penicillin allergy label and for patients with the label removed, and cost-effectiveness. We evaluated the methodological quality of the studies using a published checklist designed for systematic reviews. The review followed a narrative synthesis.

**Data availability statement:** The detail data extracted from the studies included in the review are now included in Supplementary File 4, and Tables S1-S4.

**Funding:** This work was part of the ALABAMA project funded by the UK National Institute of Health Research Programme Grants for Applied Research (RP-PG-1214-20007), which sought to conduct a randomised controlled trial and cost-effectiveness analysis of testing for penicillin allergy. The funders had no role in study design, data collection and analysis, decision to publish, or preparation of the manuscript.

**Competing interests:** The authors have declared that no competing interests exist.

## Results

Thirty-six studies met the inclusion criteria. Most studies analysed the effect of testing on the costs of antibiotic use among patients admitted to hospital with a PAL. Studies measured costs of testing (n = 19); antibiotic medication use (n = 23); adverse reactions with penicillin use (n = 4), alternative antibiotic drugs (n = 3); length of hospital stay (n = 5); subsequent health care use episodes (n = 4); and antibiotic medication use in subsequent care episodes (n = 3). The median cost of skin testing plus oral challenge across six primary costing studies was USD 246 (range: 164, 514), which contrasts with the USD 42–258 range of antibiotic cost savings during the initial hospital admission. Two studies presented evidence that penicillin allergy testing is cost-saving in an outpatient setting over 3.5–4.5 years. One model-based study reported that testing in inpatient settings is cost-saving. No reports on the effect of penicillin allergy testing on health-related quality of life were found and the two cost-effectiveness studies that accounted for this outcome employed the opinion of healthcare professional or an assumption of a common generic value for adverse reactions.

## Conclusions

While penicillin allergy testing shows promise in reducing antibiotic costs, the evidence remains insufficient to definitively establish whether these savings consistently outweigh testing costs across various healthcare settings.

## Introduction

Penicillin and other beta-lactam antibiotics are the most frequent cause of medication induced anaphylaxis [1–3] and 6% of people in England have a record of penicillin allergy [4]. However, at least 9 out of 10 people who believe they are penicillin-allergic are found not to be when tested, and so could safely take penicillins [5,6]. These patients may be receiving less effective antibiotic treatments with additional long-term health risks. Penicillin-allergy records (or label) drive prescribing towards alternative broad-spectrum antimicrobials that contribute to increased antimicrobial resistance (AMR) and may result in poorer patient outcomes. Research has found that macrolide, tetracycline, cephalosporin and quinolone prescribing were all more common in patients with a record of penicillin allergy, compared to those without, and that antimicrobial prescriptions were almost twice as frequent in patients with a penicillin allergy label [4,7].

Antibiotic allergies, including penicillin allergy, have been associated with sub-optimal antibiotic therapy, increased antimicrobial resistance, increased length of stay, increased antibiotic-related adverse effects such as *Clostridioides difficile* infection, intensive care unit (ICU) admission, death, as well as increased treatment cost [8]. Antibiotic regimens deviate from the standard of care in approximately 40% of patients who report a penicillin allergy [9,10]. The costs of the consequences of sub-optimal antibiotic therapy due to reported penicillin allergy are likely significant, reaching far beyond the actual differences in antibiotic costs [11]. Given the

significant proportion of incorrect penicillin allergy labels and their impact on healthcare costs and outcomes, it is critical to evaluate the economic implications of penicillin allergy testing.

The standard assessment for penicillin allergy involves skin testing followed by intradermal tests and oral drug provocation testing (oral challenge test). Several recent studies have reported safe and effective testing for low-risk penicillin allergy labels using a direct oral challenge in children and increasingly in adults [12–15]. Due to the prevalence of penicillin allergy labels and the limited capacity of existing specialist allergy clinics, there is increasing interest in expanding access to penicillin allergy testing through provision of non-allergy specialist delivered testing for low-risk patients. Evidence on the costs and benefits of different models of penicillin allergy testing services that helps inform policy decisions and service design and planning is limited, as existing reviews have looked at the associated costs of penicillin allergy labels rather than assessments of testing models and predate the advent of penicillin allergy testing models for low risk patients by non-allergists [16].

The aim of this study was to review the published evidence on the cost-effectiveness of penicillin allergy testing and impact on healthcare costs and health-related quality of life associated with removing a penicillin allergy label. This evidence is intended to inform sustainable service delivery models for increasing access to penicillin allergy testing and de-labelling of non-allergic patients.

## Methods

This systematic review was conducted according to the Cochrane guidelines [17], and is reported in line with PRISMA 2020 [18] (S1 File). The review protocol is registered in PROSPERO (CRD42021231848).

### Search strategy

The searches were designed and run by an information specialist in November 2020, then rerun in full on 14th November 2023 in Medline, Embase, Scopus, Web of Science, EconPapers (RePeC) and the International HTA Database (INAHTA). Text words and database subject headings were used for two search concepts, penicillin allergy and economic evaluations. The searches were not limited by date or language and were peer-reviewed by a second information specialist using the PRESS checklist [19]. See S2 File for full details of the search strategies.

### Eligibility criteria

We included all studies that had: i) patients with a recorded or self-reported penicillin allergy label; ii) used a test to delabel the penicillin allergy; iii) contained one or more comparator groups; iv) measured both costs and health outcomes (full economic evaluation) or either costs or quality of life outcomes (partial economic evaluation). Only peer-reviewed publications written in English were included.

We excluded studies that did not perform a penicillin allergy test, conference abstracts, short notes, comments, editorials and study protocols. Reviews that did not synthesise new cost effectiveness estimates were excluded; however, their reference lists were screened for additional records.

### Outcomes

The outcomes of interest were healthcare resource use, medical costs, and health-related quality of life for patients with a penicillin allergy label and for patients with the label removed. In addition, reports on cost-effectiveness of interventions for safely de-labelling individuals with an incorrect penicillin allergy label were reviewed.

### Study selection

Two reviewers independently screened the title and abstract of each record using Rayyan software [20]. Full texts of potentially relevant records were then reviewed against the eligibility criteria for study inclusion. Disagreements were resolved by discussion with a third reviewer.

## Data extraction

Study characteristics were extracted from each included study, including publication details, the type of study conducted, the population included in the study, the intervention and comparators. Any results reported on health outcomes and cost measures, total or incremental costs were also extracted. Data was independently extracted by two reviewers, any discrepancies were resolved by a third reviewer.

## Quality and risk of bias assessment

In a slight deviation from the protocol, we evaluated the methodological quality of the included economic evaluations using a published checklist designed for systematic reviews [21] instead of the CHEERS statement [22], which is a reporting checklist as opposed to a quality assessment tool. Each study was appraised independently by two reviewers, with conflicts resolved by discussion. In addition, the ROBINS-I tool was used to assess risk of bias for each healthcare cost and health outcome reported in observational studies [23].

## Data synthesis

The narrative synthesis was structured according to study setting (primary or community, hospital outpatient and inpatient, other), study design (prospective, retrospective or model based), and study type (costing study, cost of illness, cost comparison/minimisation analysis, cost effectiveness analysis). We also recorded whether the study was 1) a cohort study with concurrent or retrospective controls, 2) a before-and-after study of provider organisation policy change or implementation, or 3) an individual counterfactual analysis of case series.

The analysis was based on data as reported in the original publications, except when sufficient data were reported to calculate statistics for unreported outcomes of interest, e.g., mean cost differences on a per patient basis. Only complete case analysis was conducted; i.e., no attempt was made to impute or otherwise adjust for missing data.

Cost estimates were presented in 2024 USD, using the latest purchasing power parities [24] and the US consumer price index [25,26] for converting original published figures reported in different years and currencies.

Intervention effect on healthcare cost estimates from primary data reported by more than one independent study were converted to percentage change units and summarised in terms of minimum, median and maximum mean reported values across studies. Effect estimates for health-related quality of life and health outcomes from multiple independent primary studies were summarised in their original units using the same statistics.

## Results

### Identified records and included studies

The search produced 2,117 records, with 1,312 unique records after de-duplication. Screening the title and abstracts led to the exclusion of 1,227 records, with 79 records included for full-text screening. This yielded 35 unique articles reporting on 36 studies (unlike the rest of the articles, which reported one economic evaluation study each, one published article reported two economic evaluation studies) meeting the final inclusion criteria. Four review articles were identified [27–30]; we hand searched the references lists of these reviews but no additional articles were identified (Fig 1; see S3 File for list of articles excluded at the full-text screening stage with reasons for exclusion).

Of the 36 included studies, 28 were observational studies and 8 were model-based evaluations. The majority of the observational studies were cost impact analyses of two or more testing strategies (n = 17), followed by studies limited to measuring the costs of testing (n = 6), and cost of therapy ('cost of illness') studies (n = 5 studies). Four of the modelling studies were cost-effectiveness studies comparing costs and health outcomes of two or more testing strategies, with the remaining four comparing only costs. The descriptive summary characteristics of observational studies are presented in Table 1 and those of modelling studies in Table 2, and their reported cost measures in S1 Table.

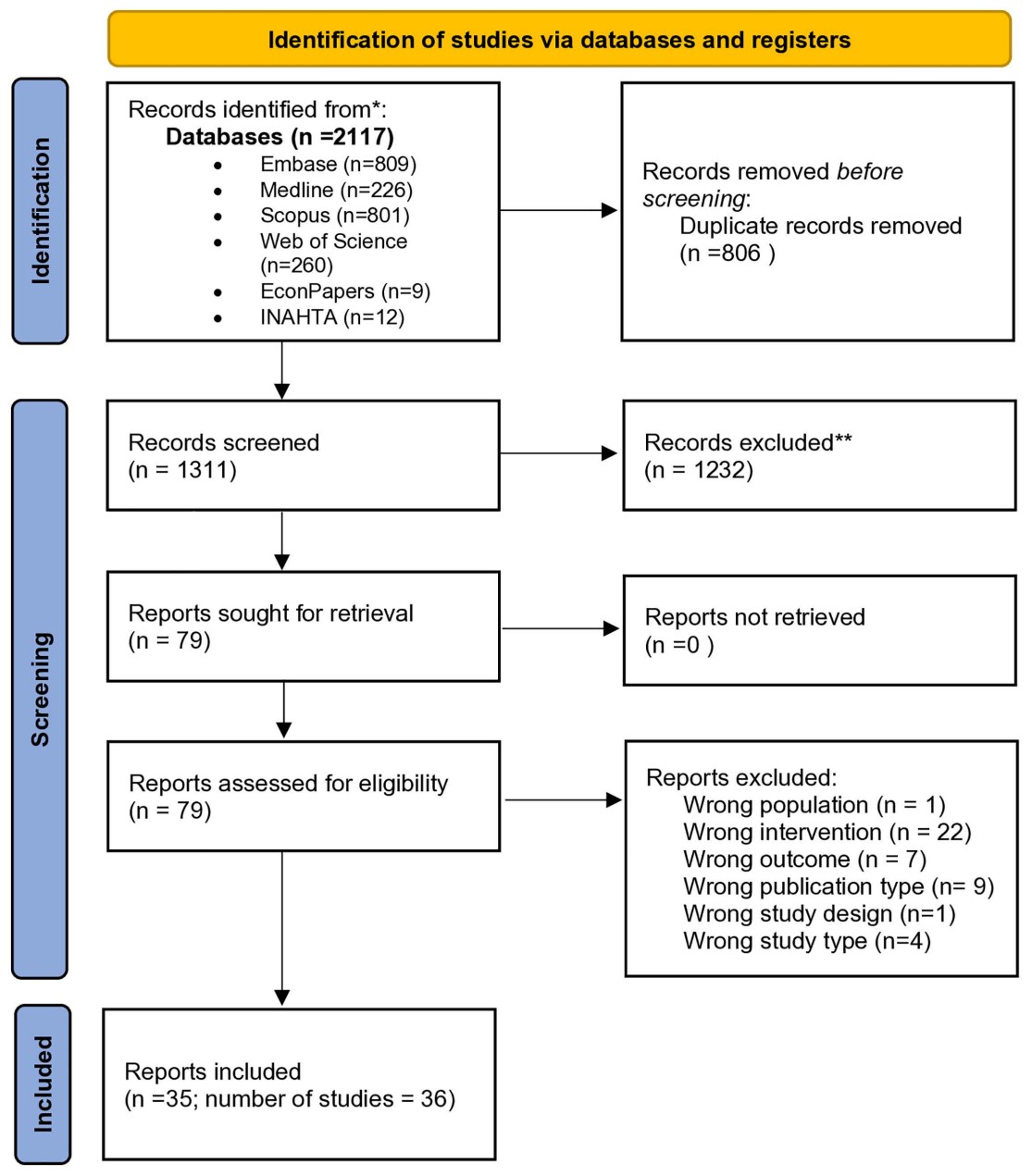

**Fig 1. Study selection.**

## Study quality

The median score across the 36 studies was 9 (range: 3–14) out of 19 items (Table 3). Whilst most studies clearly described the population, the testing strategies under comparison and their study question, 50% (n = 18) of studies adopted a time horizon that was too limited to capture relevant costs and health outcomes to their study question (e.g., until discharge for studies of inpatients or until initial antibiotic course was complete). Only 11% (n = 4) and 44% (n = 16) of studies adequately measured health outcomes and costs respectively (e.g., average antibiotic costs per day of regimen

**Table 1. Description of observational studies.**

| Authors & Date of publication | Country, currency & study date | Prospective, retrospective | Type of economic evaluation | Time horizon & discounting | Patient Population | Intervention (I) | Comparator (C) | Who is doing the test? | Outcomes measured | Perspective | Cost categories measured |
|---|---|---|---|---|---|---|---|---|---|---|---|
| | **Outpatient setting** | | | | | | | | | | |
| Allen et al. 2021 [31] | Ireland, Euro, study date not stated | Prospective | CMA | End of test | children with non-immediate symptoms of an allergic reaction outpatient | TM screening consultation + single dose OPD in OP setting & 5-day home Ab course | OP Screening consultation (I) as ward admission | OPD: non-consultant doctor (I); consultant-supervised nurse (C) | None | NHS | Costs of testing |
| Blumenthal et al. 2018 [32] | US, USD, date not stated | Prospective | Costing | End of test | Outpatients presenting for an allergy evaluation | Skin test + oral challenge | N/A | Allergy specialist | None | Health system | Costs of testing |
| Englert et al. 2019 [33] | US, USD, 2017–2018 | Prospective | Costing | Test | Outpatients with record of Type I IgE-mediated PenA prescribed alternative Ab | Skin prick test + IDT | None | Trained pharmacist | change to preferred Ab, LOS (not comparative) | Provider | Testing supplies, Ab (not comparative) |
| Ferre-Ybarz et al. 2015 [34] | Spain, Euro, 2009–2010 | Retrospective | CMA | 15 days after DPT | Patients referred to allergy department suspected of allergy to BLs | Skin test with BLs + DPT suspected & alternative drug | Direct DPT | Allergy specialist | None | Provider | Cost of testing |
| Jaoui et al. 2019 [35] | France, Euro, 2015–2018 | Retrospective | CMA | 8 days after first dose | children referred for history of mild non-immediate reactions to BL outpatient | Direct ambulatory DPTs | Direct inpatient DPTs | Not stated | None | NHS | Testing with inpatient vs outpatient visit for DPT testing |
| Macy 1998 [36] | US, USD, 1994–1995 | Retrospective | Cost of illness | One year | Health plan members (KP) who obtained a prescription medication from a health plan pharmacy | Skin test + Oral challenge | None | Allergist | Adverse reaction | Payer | Fixed Pharmacy dispensing & Antibiotic use (outpatient) |
| Macy et al. 2017 [37] | US, USD, 2010–2015 | Retrospective, observational | CMA | 3.5 years | Patients attending a hospital allergy consultation (a few as inpatient) | PST + oral challenge | No testing | Registered nurses | Failure to amend EHR (7%) | Third party payer | Testing, OPD visits, ED visits, inpatient days |
| Sobrino et al. 2020 [38] | Spain, USD, 2017–2018 | Prospective study | Costing | Until testing | Pediatric patients with suspected β-lactam allergy who consulted for allergy outpatient evaluation | SPT + IDT & patch test when advisable + ODP | None | Allergist | None | NHS & Societal | Testing (inc. patient travel costs, productivity losses by guardian) |
| Sobrino et al. 2021 [39] | Spain, Euro, 2018–2018 | Prospective observational | Costing | Until testing | Patients aged ≥14 years attending the allergy outpatient clinic for suspected HS reactions to β-lactams | SPT + IDT & patch test + ODP with various β-lactams | None | Unclear | Allergic, number of reactions | NHS & Societal | Testing (inc. patient travel costs & productivity losses) |
| Vyles et al. 2018 [40] | US, USD, date not stated | ProspectiveCase series | Costing | 1 year post testing | Children with reported penicillin allergy and low-risk symptoms in paediatric ED | skin test + graduated oral challenge | None | Unclear | Primary care record label removal PenAR | Provider | Subsequent antibiotic use |

*(Continued)*

| Authors & Date of publication | Country, currency & study date | Prospective, retrospective | Type of economic evaluation | Time horizon & discounting | Patient Population | Intervention (I) | Comparator (C) | Who is doing the test? | Outcomes measured | Perspective | Cost categories measured |
|---|---|---|---|---|---|---|---|---|---|---|---|
| **Inpatient setting** | | | | | | | | | | | |
| Borch et al. 2006 [4] | Denmark, Euro, 2003 | Prospective | CMA | End of test | Inpatient with record or self-reported penicillin allergy | Skin test+IDT+oral challenge | No testing | Not stated | None | Provider | Ab drug use in hospital |
| Brusco et al. 2023 (I) [41] | Australia, AUS$, 2019–2020 | Prospective- with matched controls | Costing | Test | Inpatients with a low risk PCN allergy label consenting to PCN allergy delabeling | Inpatient delabelling using oral challenge, direct delabelling or skin test | OP delabelling after hospital discharge | Trained nursing, pharmacy, medical staff | None | societal | Testing, travel and time of patient & carer |
| Brusco et al. 2023 (II) [41] | Australia, AUS$ 2015 & 2019 | Prospective, Before and After | CMA | Hospital Discharge | Inpatients with a low-risk PCN allergy label, and an infective diagnosis | Delabeling Program using OC, DD or ST | Usual care cohort (no testing) | Trained Nursing, pharmacy, medical staff | None | Health carer sector | Ab, total admission |
| Chen et al. 2018 [42] | US, USD, 2015–2016 | Retrospective, Historical cohort | CMA | Discharge | Inpatients with penicillin allergy being prescribed aztreonam | eHR Decision support tool for active screening (using skin test+ODP) | Active screening only (w/ skin test+ODP) | Allergy trained pharmacist with allergist support | None | Provider | Costs of testing+inpatient Ab drug use |
| Du Plessis et al. 2019 [43] | NZ, NZ$, 2015 | Prospective, Before and After | CMA | 1 yr post screening consultation | Adult Patients admitted with a penicillin allergy label | ODP | No testing | Trained pharmacist | Willingness to take penicillin | Provider | Costs of AB medication in the community |
| Fan et al. 2020 [44] | US, USD, 2015–2018 | Retrospective chart review | CMA | Hospital discharge | Breast surgery patients with a PNCA and a peri-operative PNCA referral | Allergy history+PCN skin test+OPD | No testing | Tests were carried in hospital's allergy clinic | None | Provider | Ab drug use in hospital & extended outpatient |
| Foolad et al. 2019 [45] | US, USD, 2017–2018 | Retrospective, Historical control | CMA | Duration of initial infectious episode | Immunocompromised adult cancer inpatients receiving aztreonam with self-reported prior possible IgE PC reaction | Skin test+DPT | No testing | Physician, fellow/ advanced practice providers | ADR | Provider | Ab drug use; cost of testing cited from a prior study |
| Forrest et al. 2001 [46] | Canada, Can$, 1993–1998 | Before and after | CMA | Duration of antibiotic administration | Inpatients requiring iv Ab≥7d & PC allergy history/ had received PST during hospitalization | History taking+Skin testing | No testing | Consultant allergist | None | Third party payer | iv Abs inc. material & labor, vancomycin, ST inc. consultation |
| Harmon et al. 2020 [47] | US, USD, 2017–2018 | Prospective, case series | CMA | 49-51 days of therapy | Any admitted adult patient with a reported PCA of any severity & were receiving non-optimal Ab | SPT+IDT+OPT | No testing | Pharmacist trained by allergy specialist | Allergic reactions | Provider/ patient (charges) | Testing (supplies of SPT only) and Ab use |

*(Continued)*

Table 1. (Continued)

| Authors & Date of publication | Country, currency & study date | Prospective, retrospective | Type of economic evaluation | Time horizon & discounting | Patient Population | Intervention (I) | Comparator (C) | Who is doing the test? | Outcomes measured | Perspective | Cost categories measured |
|---|---|---|---|---|---|---|---|---|---|---|---|
| Heil et al. 2016 [48] | US, USD, Date not stated | Prospective Before-and-after | CMA | Days of therapy until discharge | Inpatients with reported penicillin allergy | SPT+IDT+OC | No testing | Infectious-disease (ID) fellow with ID physician supervision | None | Provider | Ab use |
| Jones and Bland 2017 [49] | US, USD, 2014–2015 | Retrospective | cost of Illness | Duration of Ab therapy | Admitted patients on Ab with suspected penicillin allergy | Two-step skin testing with optional oral challenge | None | Nurse assisted by stewardship pharmacist | None | Provider | Antibiotic use |
| Jones et al. 2019 [50] | US, USD, 2015–2017 | Retrospective | cost of illness | Duration of Ab therapy | Admitted patients on Ab with self-reported penicillin allergy | Two-step skin testing with optional oral challenge | None | Nurse assisted by stewardship pharmacist | Proportion of Ab therapy days on BL | Provider | Testing materials and Antibiotic use |
| King et al. 2016 [51] | US, USD, 2013–2015 | Retrospective | Costing | Duration of Ab therapy | Adult inpatients with a β-lactam allergy receiving broad-spectrum Ab | PST + oral challenge | None | Allergist | Switch rate to BLA 75%, BL use subsequent hospital episode | Provider | Test supplies and Antibiotic use |
| Li et al. 2019 [52] | Australia, Aus$, 2017–2018 | Prospective case series, matched controls | CMA | 6-months after discharge | Inpatients >16 years with diagnosis of type B penicillin allergy that required penicillin-containing Ab therapy | Oral challenge (Type A, Type B & type 1/mild type 4); Skin test+oral challenge (Type B & anaphylaxis within 10 y) | No testing | Allergy/immunology registrar with allergist guidance | ICU admission, Readmissions, Multi-resistant organism acquisition | Provider | Hospital bed-days, Ab |
| Modi et al. 2019 [53] | US, USD, 2010–2019 | Retrospective, before and after | Cost of illness | Hospital discharge (90 days for outcome) | Adults, self-reported β-lactam allergy undergoing inpatient hematopoietic stem cell transplant | SPT+IDT+OPT (inpatient or outpatient) | No mandatory skin testing protocol | Consultant allergist | CDI incidence, mortality | Provider/payer | Antibiotic medications, SPT (charges) |
| Ramsey et al. 2020 [54] | US, USD, 2018–2019 | Prospective | CMA | 2 weeks | Inpatients adults receiving any antibiotic and reporting a Pen-allergy | Low-risk[1]: 3-step amoxicillin DC; High-risk[2]: PST+ amoxicillin dose | None | Allergist | AR, Ab use | Provider | Ab, testing |
| Rimawi et al. 2013 [55] | US, USD, 2012 | Retrospective, before and after | Cost of illness | Duration Ab therapy | Inpatients of any age, exclude uncertain allergy history cases | PST+IDT+OC | None | Supervised infectious disease fellow | AR | Provider | Ab |
| Staicu et al. 2018 [56] | US, USD, 2017 | Prospective | CMA | 2-4 weeks | Adult penicillin-allergic inpatients receiving systemic antibiotics | SPT+IDT+amoxicillin challenge with post-ST allergist consultation | None | Allergy/immunology physician assistant | TM utility, patient satisfaction | Provider | Ab, test supplies |

Notes: CMA: cost-minimisation analysis, TM: telemedicine, SPT Skin prick test, ST Skin test, DPT Direct Provocation test, OC Oral challenge, DC Direct oral challenge, DPT Drug provocation testing, DD Direct delabelling, ODP Oral drug provocation, IDT Intradermal test, TSA Total Shoulder Arthroplasty, TKA Total Knee Arthroplasty, BEE Break-even evaluation. [1] Low risk patients: history of cutaneous only reaction to penicillin> 20 years prior; High risk patients: history of cutaneous-only reaction<=30 years prior or history of angioedema or reactions involving multiple body systems.

**Table 2. Description of modelling studies.**

| Authors & Date of publication | Country, currency & study date | Modelling study? | Type of economic evaluation | Time horizon & discounting | Patient Population | Intervention (I) | Comparator (C) | Who is doing the test? | Outcomes measured | Perspective | Cost categories measured |
|---|---|---|---|---|---|---|---|---|---|---|---|
| Bragg et al. 2023 [57] | US, USD, date NA (assumption) | Algebraic calculation | BEA | 2 years | TSA patients reporting penicillin and cephalosporin allergies | Skin test + oral challenge | No testing | Allergy specialist | None | Provider | Testing, revision for peri-prosthetic joint infection |
| Dodek et al. 1999 [58] | Canada, Can$, date not stated | Decision tree | CUA | Hospital discharge | Endocarditis from cloxacillin-susceptible SA infection & history of immediate hyper-sensitivity to penicillin | Skin testing | No skin testing & treat with vancomycin | Allergist | QALYs | Third-party payer | Skin test, measurement of serum Ab levels, medications, hospital stay |
| Lee et al. 2021 [59] | US, USD, date NA (assumption) | Survival extrapolation | CMA, BIA | 10 years | Patients undergoing hip and knee replacement with a self-reported penA | SPT | No testing (standard of care) | Not stated | None | Not stated | Testing, prosthetic septic joint revision |
| Mattingly et al. 2019 [60] | US, USD, date not stated | Decision Tree | CUA | One Year | Adults with self-reported Penicillin allergy in an inpatient setting undergoing treatment for MSSA bacteremia | Skin test | no testing (standard of care) | Not stated | QALY, re-admission rate | Health sector | Testing, Inpatient & OP Ab use, Inpatient & OP care for MSSA, AR to therapy |
| Pagani et al. 2021 [61] | US, USD, date NA (assumption) | Algebraic calculation | CMA, BEA | 10 years | Patients reporting penicillin & cephalosporin allergies before elective THR/TKR | Penicillin skin testing plus one-step amoxicillin drug challenge | No testing | Allergist/immunologist | None | Provider | Testing, prosthetic joint infection revision |
| Phillips et al. 2000 [62] | Canada, Can$, date not stated | Decision Tree | CEA | Hospital discharge | Patients with PenA label Ab undergoing cardiovascular surgery with Ab prophylaxis | Skin testing strategies: 1) Triaged from oral history 2) Add-on to oral history 2) Test all PenA labelled | Two options: 1) No testing 2) Oral history suggesting IgE-mediated reaction to Pen | Not stated, costing is from drug safety clinic | Reaction avoided: Anaphylaxis (shock, RD); SAR (hives/rash) within 72 h. | Provider | Ab (acquisition and delivery), skin testing, AE (LOS) |
| Sousa-Pinto et al. 2021 [63] | US & Europe, USD, 2018 | Decision tree | CMA | 30 days post first discharge (inpatients); 5 years post testing (outpatients) | Patients with PenA label in different settings, in/ outpatient, geographies | 1) ST + ODP 2) ODP alone | No testing – all treated as allergic | Unclear – seem allergy clinic | None (severe reactions costed) | Health service | Testing, Hospital bed-days, outpatient visits, Ab use, allergy medication |
| Thao et al. 2023 [64] | US, USD, date not stated | Decision Tree | CEA | Until delivery | Pregnant women with self-reported PenA receiving intrapartum Group B streptococcus prophylaxis | PST + OC | No testing (usual care) | Allergy specialist | Appropriate Ab use | Third-party payer (Medicare & Medicaid) | Testing, Ab use |

BEA Break-even analysis, BIA Budget Impact Analysis, CEA Cost-effectiveness Analysis, CMA Cost-minimisation Analysis, CUA Cost utility analysis, Ab Antibiotic, AE Adverse event, LOS, Length of hospital stay, RD Respiratory distress, OP outpatient, TSA NA Not applicable (test evaluation based on assumption).

**Table 3. Study Quality: CHEC checklist (Evers et al. date).**

| Item | % studies (N = 36) meeting the item criteria |
|---|---|
| 1. Is the study population clearly described? | 100 |
| 2. Are competing alternatives clearly described? | 89 |
| 3. Is a well-defined research question posed in answerable form? | 97 |
| 4. Is the economic study design appropriate to the stated objective? | 72 |
| 5. Is the chosen time horizon appropriate in order to include relevant costs and consequences? | 50 |
| 6. Is the actual perspective chosen appropriate? | 69 |
| 7. Are all important and relevant costs for each alternative identified? | 44 |
| 8. Are all costs measured appropriately in physical units? | 39 |
| 9. Are costs valued appropriately? | 58 |
| 10. Are all important and relevant outcomes for each alternative identified? | 11 |
| 11. Are all outcomes measured appropriately? | 8 |
| 12. Are outcomes valued appropriately? | 0 |
| 13. Is an incremental analysis of costs and outcomes of alternatives performed? | 67 |
| 14. Are all future costs and outcomes discounted appropriately? | 19 |
| 15. Are all important variables, whose values are uncertain, appropriately subjected to sensitivity analysis? | 25 |
| 16. Do the conclusions follow from the data reported? | 75 |
| 17. Does the study discuss the generalizability of the results to other settings and patient/client groups? | 64 |
| 18. Does the article indicate that there is no potential conflict of interest of study researcher(s) and funder(s)? | 39 |
| 19. Are ethical and distributional issues discussed appropriately? | 3 |

were reported as opposed to average per patient costs over a defined follow-up length), and 67% (n = 24) clearly presented an analysis of costs and/or benefit differences. In 75% (n = 27) of studies, no sensitivity analyses of assumptions used in measuring or analysing costs or benefits were presented, and 36% (n = 13) of studies did not discuss the generalisability of their findings beyond their local study setting. Almost two-thirds of studies reported conflicts of interest or presented no information on whether such conflict existed (Table 3).

## Costs measured in the studies

**Costs of evaluating Beta-lactam allergy.** Nine studies reported on the costs of testing, of which six evaluated skin tests with oral challenge (ST + OC) [32,34,38,39,42,54](apart from two [32,42], these studies used intradermal injection after negative skin prick testing), and six evaluated direct drug provocation (DPC) testing [31,32,34,35,41,54]. The median cost of ST with oral challenge at an outpatient allergy clinic was USD 246 (range: 164, 514), whilst DPC in an outpatient allergy clinic was USD149 (range: 71, 253; S4 File). A prospective study in children reported that the cost of DPC delivered by a nurse and supervised by a consultant amounted to USD 800 [31]. A second report of DPC delivered by non-allergy specialists reported costs of USD 172 for low-risk patients tested by trained nursing, pharmacy or medical staff in Australia [41] (Table 4; S4 Table).

Three studies investigated patient travel and work absenteeism costs associated with seeking and receiving testing, the costs of which were as large as those of testing itself to the health system [38,39,41]. Two of these studies also reported variation in costs according to test result, which was negligible. They also reported a cost comparison between patients with immediate and patients with delayed reactions, with the latter being 50% larger than the former in adults [39] and negligible in children [38].

Low-risk penicillin allergy de-labelling in an inpatient setting was found to cost less than attending an outpatient clinic following discharge from hospital, whether only the direct healthcare costs or also patient and carer travel costs were

**Table 4. Direct cost of PAL testing.**

| Testing strategy | Number of studies | Cost per patient (USD*) | | | Comments and sources |
|---|---|---|---|---|---|
| | | Median of study means | Minimum of study means | Maximum of study means | |
| Skin Test (outpatient) | 1 | 190.57 | 190.57 | 190.57 | Trained nurse, pharmacist or medical staff [41]; skin prick and intradermal testing |
| Sink Test (inpatient) | 1 | 453.20 | 453.20 | 453.20 | Nurse, pharmacist or medical staff [41]; skin prick and intradermal testing |
| Skin test plus oral challenge (outpatient) | 6 | 245.78 | 164.32 | 514.38 | Allergist [32,34,38,39,41,54]; one was a retrospective study [34]; skin prick test + IDT, with one exception that did not use IDT [33][a] |
| Skin test plus oral challenge (inpatient) | 1 | 287.02 | 287.02 | 287.02 | Pharmacist [42](skin prick test without IDT) |
| Oral Challenge (non-allergy specialists) | 2 | 485.91 | 171.57 | 800.25 | Outpatient. Trained nurse, pharmacist or medical staff [41]; consultant-supervised nurse [31] [b] |
| Oral Challenge (allergist) | 4 | 148.77 | 70.57 | 252.73 | Outpatient [32,34,35,54]; two retrospective [34,35] [c] |

*At year 2024 prices. [a]No study reported measure of sampling uncertainty [b]No study reported measure of sampling uncertainty. [c]No study reported measure of sampling uncertainty. IDT: Intradermal test.

included in the analysis [41]. This was due to the high proportion of patients missing their de-labelling appointments (11%) and having separate visits for assessment and testing (17%) in the outpatient setting, whereas inpatients had their allergy assessment, testing and communication of results within the index admission without needing further clinical appointments.

**Costs of antibiotic medication use.** Antibiotic costs were the most commonly investigated outcome (reported in 19 of the included studies), with all studies documenting cost savings [7,33,36,40–47,49–56]. The ability to compare their outcomes is low as studies varied widely in their methodology. In cohort studies of the index admission with concurrent or historical controls [41,42,52], the median reduction of antibiotic costs associated with testing was 57% (range: 53,66), and 52% (range 20,66) when extending the study set to include uncontrolled cohort studies [43,46] (Table 5; S4 File). Per patient antibiotic cost savings in these studies, excluding the study in aztreonam users [42], ranged between USD 42–258 (S3 Table).

In contrast, uncontrolled studies of 'potential impact' compared the observed antibiotic costs in a group of inpatients undergoing penicillin allergy testing with the counterfactual costs that would have been observed if those patients had not had their PAL tested [7,51,55,56]. The median reported projected reduction in antibiotic costs with skin testing plus oral challenge in the inpatient setting [7,51,55] was 57% (range: 29, 83).

Three studies [36,40,43] reported the effects on costs of antibiotic use in the community over a 1-year period after testing, resulting in a median reduction of 52% (range 42,66). One study reported antibiotic cost savings per patient of USD 22 associated with removing a PAL in 81 low-risk children attending the emergency department [40]. A second study compared the total number of antibiotic courses in the year before with the year after intervention in 250 patients at the imputed cost of the preferred prescribed antibiotic, which resulted in an antibiotic cost per day per patient that was 2.5 times greater in confirmed penicillin allergic patients than de-labelled patients [43]. A third study in 236 patient who obtained a prescription medication from a health plan pharmacy reported USD 48 in cost savings [36].

**Costs of healthcare resource utilisation.** Four studies measured the impact of penicillin allergy de-labelling on hospital length of stay (LOS), three were limited to initial admission LOS [33,41,53] and one looked at LOS for admissions up to 6 months after testing [52]; two of these reported sufficient information to estimate the effect on costs [41,52]. Adult inpatients with a type B penicillin allergy diagnosis requiring penicillin antibiotic treatment who were tested with DPC

**Table 5. Effect on healthcare costs of PAL testing (USD).**

| Healthcare resource | Number of studies | % reduction in cost per patient | | | Risk of bias[a] | Comments and sources |
|---|---|---|---|---|---|---|
| | | Median of study mean differences | Minimum study mean | Maximum study mean | | |
| Antibiotic use -initial therapy | 5 | 53 | 20 | 66 | Moderate-Severe | Median is clinical support tool intervention promoting PADL by pharmacist [42]; minimum is from study of Guideline intervention study for PST performed by allergist [46]; maximum is from a study of inpatients until hospital discharge [52]; other are prospective study of inpatients until hospital discharge [41] and study of inpatients whose result was not reported but derived from reported median LOS times drug related costs per inpatient in each arm [43] |
| Hospital LOS (Initial hospital admission) | 2 | 35 | 23 | 47 | Moderate | Median: mid-point value of minimum and maximum. Minimum is from study report that DPC or SPT+OC reduced LOS by 23% among adult patients admitted with a diagnosis of type B Penicillin allergy that required penicillin-containing antibiotics (p<0.05 [52]); maximum is from study report that DPC among low risk admitted adults reduced such costs by 47% (p=0.002 [41]). |
| Antibiotic use (Community) | 3 | 52 | 42 | 66 | Severe | Median: estimate from an outpatient cohort over a 1 year follow up period (20% loss to follow-up) after testing including fixed pharmacy dispensing costs [36]; minimum is from a 1 year follow-up study in children [40]; Maximum is from admitted adult patients with a PAL over 1 year follow-up; derived from published study data on cost per day of regimen and number of regimens and assumption of 5-day regimen duration [43]. |

[a]ROBINS-I tool V2 [23]. PAL: Penicillin allergy label. SPT: Skin prick test. DPC: Direct penicillin challenge. OC: Oral challenge. PADL: Penicillin allergy delabelling.

or ST+OC had a reduced LOS (23%, p<0.05) [52], whilst the costs of admission (emergency department and acute including antibiotics) were reduced by 47% (p=0.002) among admitted adults who were low-risk and had a DPC [41]. The respective cost savings of these studies were USD 4602 and 8012 (95% CI: 13,050–2975; S4 Table). One of the studies that presented no cost data found no statistically detectable differences in LOS during index hospitalisation nor in length of ICU stay [53].

In a retrospective study of insurance claim records, individuals with a penicillin allergy history attending an outpatient allergy consultation for skin testing plus an oral challenge were subsequently observed to experience 0.55 fewer days in hospital and 0.09 fewer outpatient visits than matched controls annually over 3.6 years [37]. In an economic model of outpatient penicillin allergy testing [63], downstream cost savings offset testing costs, and were driven by an effect of de-labelling of 3.05 (95% CI: 2.3, 3.8) fewer annual primary care visits over 4.5 years. This value was obtained from a published estimate of the excess number of primary care visits in an observational study of a group of patients with a PAL relative to matched controls without a PAL [65]. The estimated minimum de-labelling rate for skin testing with oral challenge to be cost-saving was 30–60% for inpatient and 10–50% for outpatient testing settings, and larger than for DPC (15–30% and 5–30%, respectively [63]).

Two publications [46,54] (three evaluated service models) reported the costs of PAL testing and antibiotic costs but lacked a control or comparable group and were based on hypothetical savings, resulting in a median net costs of USD 39 (range, −39, 167). One matched controlled study [37] reported sufficient data to calculate the costs of testing for PAL and healthcare use over a 3.6 follow-up period after testing, resulting in cost savings of USD 8,811 (Table 6).

**Health outcomes.** Two studies reported health outcomes [52,53]. In a retrospective study of adults undergoing inpatient hematopoietic stem cell transplantation in a single centre, no detectable difference in 90-day post-transplant incidence of *Clostridioides difficile* nor mortality was observed (11% before and 7% after, p=0.34, in both outcomes) after

**Table 6. Cost difference in PAL testing and antibiotics costs between testing and not testing for PAL.**

| Healthcare resource | Number of studies | Cost difference (USD)[a] | | | Risk of bias[b] | Comments and sources |
|---|---|---|---|---|---|---|
| | | Median of study mean cost differences | Minimum | Maximum | | |
| Penicillin Tests & Ab use (index hospital admision) | 2 | 31.83 | −39.25 (SD 34.88) | 167.27 | Critical | Median: net costs reported by study of OC test performed by allergist [54]; Minimum: Guideline intervention study for PST performed by allergist [46]; Maximum: SPT by allergist in outpatient setting [54] |
| Penicillin test & OP | 1 | −59.36 | −59.36 | −59.36 | Moderate | Outpatient and a few inpatient tests, over 3.6 years follow-up, P < 0.001 [37] |
| Penicillin test & OP & ED | 1 | −808.06 | −808.06 | −808.06 | Moderate | Outpatient and a few inpatient tests, over 3.6 years follow-up [37] |
| Penicillin test & OP & ED & inpatient days | 1 | −8811.03 | −8811.03 | −8811.03 | Moderate | Outpatient and a few inpatient tests, over 3.6 years follow-up [37] |

[a]At US prices of 2024. [b]ROBINS-I tool V2 [23]. PAL: Penicillin allergy label. SD Standard deviation. Ab: Antibiotic. SPT: Skin prick test.

mandatory implementation of ST + OC [53](Risk of bias: Severe, before-cohort vs after-cohort study without concurrent control). In a prospective case series study of 70 ST + OC tested adult inpatients requiring penicillin, their rate of 6-month hospital readmission was 43% versus 61% in matched controls (p < 0.05), and the rate of newly acquired multi-resistant organisms was 4 and 7% (p = 0.50), respectively [52](Risk of bias: Moderate, cohort study with concurrently controls).

**Health-related Quality of life.** No study was found that reported evidence on the health-related quality of life of individuals with a PAL. Only a couple of studies, both based on decision tree models, reported results in terms of quality-adjusted life years (QALYs). One of them elicited health state utility values of treatment cure and failure in *Staphylococcus aureus* endocarditis under skin testing with and without adverse reaction to penicillin or toxicity from second-line therapy (vancomycin), and under no testing with and without vancomycin toxicity, from a nurse providing care for such inpatients [58]. The other was a study of skin testing in *S. aureus* bacteremia which obtained values for a 'post-septic episode with no other issue', 'disutility for adverse event/adverse drug reaction' independent of antimicrobial regimen and 'disutility for readmission' from previous studies in patients with bacteremia [60].

**Cost-effectiveness of de-labelling individuals with an incorrect penicillin allergy label.** Four studies reported results combining differences in costs and health benefits in cost-effectiveness analyses, all of them using decision trees. One study compared skin prick testing (SPT) with no SPT over a 1-year time horizon, including the inpatient and outpatient costs of antibiotic therapy and care for complicated infections and health-related quality of life losses associated with readmissions and adverse events [60]. A second study modelled outcomes of SPT-guided management of patients with *S. aureus* infective endocarditis followed up until hospital discharge and found it to have lower expected costs and higher health-related quality of life (utilities) than the alternative of not testing and treating patients with vancomycin [58]. In contrast, SPT-guided vancomycin prophylaxis for cardiovascular surgery patients with a PAL had incremental costs per anaphylactic case avoided above USD 400,000 and were not recommended for routine use instead of oral history-based management [62]. The fourth study evaluated ST + OC-guided intrapartum Group B Streptococcus (GBS) prophylaxis in pregnant women, in which the costs were included for penicillin allergy and GBS sensitivity testing and antibiotics, resulting in a cost per additional patient appropriately treated of USD 1360 relative to usual treatment according to the PAL [64].

## Discussion

Of the 36 studies included in this review, the majority met less than half the methodology quality appraisal items required to be considered a robust cost-effectiveness analysis. Most were cost analysis studies, focusing on antibiotic and direct

healthcare costs before and after testing. Only a few studies evaluated the impact of testing and de-labelling outside a hospital setting, and most of these included small inpatient populations, were confined to single specialties (e.g., joint replacement surgery [61], and breast surgery [44]), and typically conducted in a single centre. Most studies were conducted over a short period of time with only a few looking at impacts beyond a 1-year time period.

This review highlights several potential economic benefits of penicillin allergy testing, primarily its impact on antibiotic acquisition cost during the index episode of care. Three studies explored the impact of allergy testing on LOS, and two report associated cost reductions to the healthcare sector of 23–48% due to LOS reductions over an acute admission alone or including repeat episodes over a 6 months period after testing. If confirmed by independent studies, this magnitude of effects on costs may offset the costs of the test, especially among low-risk patients timely screened for DPC after admission. There were no studies that reported on the patient benefits of an earlier discharge from hospital, e.g., due to switching from complex antibiotics to simpler oral penicillin administration [49,55].

A couple of studies documented the cost of immediate and delayed reactions to the test, but there were no studies reporting the economic consequences of any potential reduction in the incidence of adverse events associated with using narrower spectrum agents due to de-labelling, e.g., C. *difficile* infection. The impact of penicillin allergy delabelling on antimicrobial acquisition costs in the community was measured in three studies; one followed children for a median of 12 months after being de-labelled at the ED [40]; another recorded costs of adult inpatients 12 months after testing [43]; and a third followed an outpatient cohort of health plan members over a 1-year period (20% loss to follow-up) after testing [36]. Neither the costs of any downstream health care service use for subsequent or relapsing episodes of infection, nor their associated health impacts were measured.

The median cost of skin testing plus oral challenge across six primary costing studies was USD 246 (range: 164, 514), which was more than the associated median antibiotic acquisition cost reductions during the index hospital episode of care (i.e. USD 74, range: USD 42–258, excluding a study in the highly selective population of patients on aztreonam [42]). This finding is also evident in the three studies that have measured the cost of testing and antibiotic use during the index hospital admission. Our study also summarises the reported unit costs of different testing strategies according to whether testing is led by an allergy specialist or non-allergy specialists in an inpatient or outpatient setting. Our review reflects the currently evolving recognition of the safety and feasibility of providing non-allergy specialist testing for penicillin allergy using DPC led by trained nurses or pharmacists [66]. Although the information presented in published studies did not always allow us to determine whether tested patients were of low, moderate or high risk, we found several reports involving high-risk populations, for example those undergoing hematopoietic stem cell transplant or the immunocompromised, where SPT and oral challenge was done by allergists. Staicu et al. [56] reported a potential saving of $152 if the testing was done by a non-allergist. A multi-test strategy of skin test plus an oral challenge was used in 18 out of 27 testing strategies, with nine of those having a three-step process (skin test, intradermal test, and an oral challenge). Skin testing alone was used in one study, and some studies estimated the cost of testing being carried out by non-allergists [31,41]. While the summarised unit costs of DPC are lower than skin test plus an oral challenge overall, there is one report of low-risk patients tested by nurse-led DPC plus remote doctor consultation (telemedicine) at a cost of USD 747, which is above most of the other estimates of skin test plus an oral challenge reported by primary costing studies and suggests results driven by differences in costing methodolgy rather than true costs.

Our study adds to a previous systematic review of the costs associated with a self-reported penicillin allergy label [16], by focusing instead on studies evaluating the effect of penicillin allergy testing interventions. Like the previous review, we found that the majority of studies are observational in nature and focused on inpatient populations. Our study adds new evidence that de-labelling may reduce costs of hospital stay during the index episode of care, but the magnitude of this effect may still not fully offset the costs of penicillin allergy testing and studies that track outcomes over 3.5 years after testing may be required to capture important cost savings from hospital stays and outpatient attendances [37].

The heterogeneous findings across studies regarding the effect of penicillin allergy testing on the costs of antibiotic use partly reflects the varying quality in this literature. For example, in measuring the impact of testing, study designs vary from measuring antibiotic use in uncontrolled case series of tested patients or controlled tested cohorts. Several studies report the difference between the costs of the observed antibiotic use in de-labelled patients and the 'theoretical' amount of antibiotic that would have been used by these patients had they retained their PAL (i.e., cost avoidance studies). Furthermore, some of these only measured costs for the subset of patients who are switched to a preferred penicillin or beta-lactam after testing as opposed to the whole sample of patients undergoing testing [45,49].

We found a number of key areas for future research. In addition to health-related quality of life, there is lack of evidence on long-term benefits of appropriate de-labelling to patients from avoiding delayed treatment of serious infections, e.g., sepsis or meningococcal meningitis. Further, none of the studies identified in this review sought to account for the population health benefits of penicillin allergy delabelling from reducing antimicrobial resistance; those that recorded antibiotic prescriptions over a one year or longer follow-up period post-testing found results consistent with reductions of 6% (p = 0.5 [43]), 14% (p not available [37]), and 28% (p = 0.0001 [36]), which suggest such potential benefits. Measurement of costs associated with rare events leading to ICU care was almost absent from the reviewed studies, reflecting the single-site nature of most of them. Multicentre controlled studies are needed to generate evidence on or adequate surrogates of these key outcomes, which would then serve to inform cost-effectiveness and cost-utility studies that account for outcomes relevant to patients as opposed to single provider institutions as in most of the received literature.

There are limitations of this review. First, we did not consider studies published in languages other than English. In view of the growing awareness of the impact of penicillin allergy labels on patient and public health we are likely to miss important emerging international evidence. Second, we excluded studies that reported quantities of resource use, such as length of hospital say, without providing the associated costs. Future reviews may seek to expand the criteria for inclusion to non-economic studies that report key outcomes, such as length of hospital stays, and apply a notional unit cost to derive more precise and informative measures of economic impact than presented here. Third, there is heterogeneity in reporting and costing methods used across studies, with some studies of ST only costing the test kit, whilst other studies accounted for the costs of staff performing the test, test kits and materials for in vivo and in vitro (intradermal) tests, cost of allergist and consultant time and capital costs of using the consultation room. We have mitigated against biases by excluding studies that did not account for the costs of consultation but observed variation across studies that may reflect differences in methods as opposed to variation in a consistent measure of unit cost of skin testing and an oral challenge or DPC test.

Despite the limitations, the reviewed evidence suggests that the economic case for penicillin allergy delabelling may be better made by carefully selecting patients on high cost antibiotics, such as admitted patients on aztreonam. Similarly, outpatients may be selected for testing among frequent users of antibiotics.

Our study findings are also consistent with the view that efficient service delivery models that rely on trained non-allergy specialists are required for the adoption of penicillin allergy testing to appeal to service managers and payers. Such adoption would be easier and less costly for hospitals with robust electronic health record systems, telemedicine services and established networks with allergy testing centres. Whilst pharmacists may have the training most suitable to lead in this role of expanding access to testing services, they would cost more than suitably trained nurses or healthcare assistants.

## Conclusions

There is limited evidence that penicillin allergy testing results in antibiotic costs savings, as published studies are observational, often uncontrolled and poorly reported. Whilst there is evidence of reduced antibiotic consumption in the community, little is known about the associated cost impact of de-labelling. Penicillin allergy testing is unlikely to recoup its costs within the first year after testing. There is emerging observational evidence to suggest significant cost savings to the health care system from reduced outpatient attendances and inpatient stays three to four years after testing. No

evidence exists on the health-related quality of life impact of penicillin allergy testing. In order to identify optimal penicillin allergy testing models and prove cost-effectiveness, randomised controlled trials with sufficiently long follow-ups and power to detect meaningful impacts to patients and national health services are urgently required, particularly in high risk and resource-constrained settings. Non-allergy specialist delivery models may offer an affordable way to expand allergy testing service beyond the limited capacity of allergy testing centres.

## Supporting information

**S1 File. PRISMA checklist.**
(DOCX)

**S2 File. Search strategy.**
(DOCX)

**S3 File. Studies excluded at full text screening stage with reasons.**
(DOCX)

**S4 File. Extracted data.**
(XLSX)

**S1 Table. Cost elements in reviewed studies.**
(DOCX)

**S2 Table. Data extraction of costs of testing.**
(DOCX)

**S3 Table. Data extraction of antibiotic costs.**
(DOCX)

**S4 Table. Data extraction of hospital costs.**
(DOCX)

## Acknowledgments

We want to thank Declan Kohl for thorough and diligent research assistance. **Declarations:** This work was part of the ALABAMA project, which sought to conduct a randomised controlled trial and cost-effectiveness analysis of testing for penicillin allergy.

## Author contributions

**Conceptualization:** Bethany Shinkins.

**Data curation:** Ruben E Mujica-Mota, Miaoqing Yang, Natalie King, Shadia Ahmed, Neil Powell, Bethany Shinkins.

**Formal analysis:** Ruben E Mujica-Mota.

**Funding acquisition:** Sue Pavitt, Jonathan AT Sandoe.

**Investigation:** Ruben E Mujica-Mota.

**Methodology:** Natalie King.

**Supervision:** Jonathan AT Sandoe.

**Visualization:** Ruben E Mujica-Mota.

**Writing – original draft:** Ruben E Mujica-Mota.

**Writing – review & editing:** Ruben E Mujica-Mota, Miaoqing Yang, Natalie King, Shadia Ahmed, Neil Powell, Sue Pavitt, Bethany Shinkins, Jonathan AT Sandoe.

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
