## [Decision Letter · Decision Letter 0]

3 Dec 2024

Dear Dr. Mujica-Mota,

Thank you for submitting your manuscript to PLOS ONE. After careful consideration, we feel that it has merit but does not fully meet PLOS ONE’s publication criteria as it currently stands. Therefore, we invite you to submit a revised version of the manuscript that addresses the points raised during the review process.

To enhance the manuscript's quality:

Focus on aligning the results and discussion with the study objectives.Address gaps in methodology and ensure clear, consistent data presentation.Strengthen the discussion by connecting findings to broader implications, such as AMR reduction and policy recommendations.

We look forward to receiving your revised manuscript.

Kind regards,

Muhammad Shahzad Aslam, Ph.D.,M.Phil., Pharm-D

Academic Editor

PLOS ONE

Journal Requirements:

3. Thank you for stating the following financial disclosure: “NIHR PGfAR RP-PG-1214-20007”.

4. In the online submission form, you indicated that “The detail data extracted from the studies included in the review are available from the authors upon request”.

All PLOS journals now require all data underlying the findings described in their manuscript to be freely available to other researchers, either 1. In a public repository, 2. Within the manuscript itself, or 3. Uploaded as supplementary information. This policy applies to all data except where public deposition would breach compliance with the protocol approved by your research ethics board. If your data cannot be made publicly available for ethical or legal reasons (e.g., public availability would compromise patient privacy), please explain your reasons on resubmission and your exemption request will be escalated for approval.

5. As required by our policy on Data Availability, please ensure your manuscript or supplementary information includes the following: A numbered table of all studies identified in the literature search, including those that were excluded from the analyses. For every excluded study, the table should list the reason(s) for exclusion. If any of the included studies are unpublished, include a link (URL) to the primary source or detailed information about how the content can be accessed. A table of all data extracted from the primary research sources for the systematic review and/or meta-analysis. The table must include the following information for each study: Name of data extractors and date of data extraction Confirmation that the study was eligible to be included in the review. All data extracted from each study for the reported systematic review and/or meta-analysis that would be needed to replicate your analyses. If data or supporting information were obtained from another source (e.g. correspondence with the author of the original research article), please provide the source of data and dates on which the data/information were obtained by your research group. If applicable for your analysis, a table showing the completed risk of bias and quality/certainty assessments for each study or outcome. Please ensure this is provided for each domain or parameter assessed. For example, if you used the Cochrane risk-of-bias tool for randomized trials, provide answers to each of the signalling questions for each study. If you used GRADE to assess certainty of evidence, provide judgements about each of the quality of evidence factor. This should be provided for each outcome. An explanation of how missing data were handled. This information can be included in the main text, supplementary information, or relevant data repository. Please note that providing these underlying data is a requirement for publication in this journal, and if these data are not provided your manuscript might be rejected.

Additional Editor Comments:

Key Strengths:

Demonstrates a standard systematic review using PRISMA.

Highlights the gap in literature, particularly in the methodological quality of primary studies.

Contributes to antimicrobial stewardship by emphasizing the role of penicillin allergy testing in rational antibiotic use, cost reduction, and quality of life improvement.

Areas for Improvement:

Abstract:

Results in the abstract are misaligned with the study's goal.

Clarify the cost categories, quality of life measurements, and effectiveness of costs examined.

Methodology:

Justify the lack of year restrictions and rationale for using a specific search timeline (2020-2023).

Provide an explanation for using 2022 USD instead of 2023 USD for cost estimates.

Include more details on the risk of bias (RoB) assessment and ensure results are visualized in the manuscript.

Explain the rationale for selecting a published checklist over the CHEERS statement and the key domains it addresses.

Clarify the conflict resolution process during data extraction and analysis.

Data Presentation:

Display study characteristics (authors, year, location, design) in the results section.

Provide statistical measures (e.g., OR, RR, mean difference) where applicable.

Address heterogeneity sources, such as variations in testing implementation or healthcare systems.

Consider including proxy models for quality of life evaluation.

Enhance visual presentation with graphs like flow charts, histograms, or comparison tables.

Discussion:

Expand on cost-effectiveness findings, including implications for non-allergy specialists conducting tests.

Address potential contributions of de-labeling to reducing antimicrobial resistance (AMR).

Discuss policy implications and propose actionable strategies for healthcare personnel.

Quantify savings from de-labeling compared to testing costs.

Suggest research designs to address gaps, such as cost-utility analyses and multicenter trials.

Conclusions:

Strengthen conclusions by aligning them with study goals and presenting evidence-based recommendations.

Highlight practical strategies to address penicillin allergy mislabeling.

Article Language and Structure:

Revise typographical errors and ensure consistent citation formatting.

Additional Comments:

Specify the median calculation approach in Table 3 and ensure consistency across strategies.

Integrate supplementary data (Tables S1 and S2) into the manuscript for better context.

Include the total number of articles reviewed in the abstract and results.

Reviewers' comments:

Reviewer's Responses to Questions

**Comments to the Author**

1. Is the manuscript technically sound, and do the data support the conclusions?

Reviewer #1: Yes

Reviewer #2: Yes

Reviewer #3: Yes

Reviewer #4: Partly

Reviewer #5: Partly

Reviewer #6: Partly

Reviewer #7: Partly

2. Has the statistical analysis been performed appropriately and rigorously?

Reviewer #1: Yes

Reviewer #2: N/A

Reviewer #3: Yes

Reviewer #4: N/A

Reviewer #5: Yes

Reviewer #6: N/A

Reviewer #7: N/A

3. Have the authors made all data underlying the findings in their manuscript fully available?

Reviewer #1: Yes

Reviewer #2: Yes

Reviewer #3: Yes

Reviewer #4: Yes

Reviewer #5: Yes

Reviewer #6: Yes

Reviewer #7: Yes

4. Is the manuscript presented in an intelligible fashion and written in standard English?

Reviewer #1: Yes

Reviewer #2: Yes

Reviewer #3: Yes

Reviewer #4: Yes

Reviewer #5: Yes

Reviewer #6: Yes

Reviewer #7: Yes

Reviewer #1: The manuscript titled "The cost-effectiveness of penicillin allergy testing: a systematic review" is well-organized, but it requires some revisions to improve clarity, consistency, and presentation. Below are my detailed peer review comments:

Title and Abstract:

Title: The title reflects the content accurately, though it may benefit from a clearer focus on the main findings. For example, consider refining it to “Cost-effectiveness of Penicillin Allergy Testing: Evidence and Gaps from a Systematic Review.”

Abstract: The abstract is concise and informative. However, the conclusion seems cautious. You might consider rephrasing it for clarity: "While penicillin allergy testing shows promise in reducing antibiotic costs, the evidence remains insufficient to definitively establish whether these savings consistently outweigh testing costs across various healthcare settings."

Introduction:

The introduction sets the stage well, presenting the problem of mislabelled penicillin allergies and the broader implications on antimicrobial resistance and healthcare costs. However, the flow could be smoother with more explicit connection between the paragraphs.

Suggestions: Add a brief transition between the discussion of penicillin allergy labels and the justification for the review. For example: “Given the significant proportion of incorrect penicillin allergy labels and their impact on healthcare costs and outcomes, it is critical to evaluate the economic implications of penicillin allergy testing."

Methods:

The systematic review methodology is comprehensive, but the presentation of search strategy and inclusion criteria might be condensed for better readability.

Suggestions: Move some of the detailed database search strategies to supplementary material and provide a succinct summary in the main text.

Risk of Bias: The explanation of bias assessment is adequate but would benefit from a brief justification for why the chosen method was more appropriate than alternatives like the CHEERS checklist.

Results:

Study Selection: The explanation of study inclusion and exclusion is clear, but the sentence on "disagreement resolution by discussion" could be more explicit about how disagreements were resolved.

Table Presentation: The tables summarizing cost data and quality of studies are highly informative but could benefit from clearer labeling. For example, some readers may not immediately understand the abbreviations (e.g., ST, OC). Including a brief key for all abbreviations would enhance clarity.

Quality of Studies: You rightly emphasize the limitations in study quality. This section could be expanded with a few examples of common methodological weaknesses in these studies to help the reader understand the issues more deeply.

Discussion:

Key Findings: The discussion appropriately highlights the variability in study quality and the evidence supporting cost-effectiveness. However, the emphasis on heterogeneity in findings may give the impression that there are few actionable conclusions. Consider highlighting which settings or populations most consistently benefit from testing.

Suggestions: Add a section that discusses practical recommendations or implications for clinicians and policymakers based on the findings.

The limitations of the review are well-articulated. However, the suggestion for future research could be more specific. Instead of general calls for more robust studies, suggest specific design improvements, such as longer follow-up periods or multicenter trials.

Conclusions:

The conclusion is balanced but somewhat passive. Consider a more active call to action, e.g., "Healthcare systems should prioritize further trials with robust design to confirm the cost-effectiveness of penicillin allergy testing, particularly in high-risk and resource-constrained settings."

General Writing and Style:

The manuscript is clear but dense in some areas, particularly in the methods and results sections. Simplifying sentences and using bullet points where appropriate (e.g., for listing databases) would improve readability.

References: The reference list is comprehensive. However, check for consistent formatting, especially the use of journal titles (some entries appear inconsistent with capitalization norms).

Figures and Tables:

Figure 1 (Study Selection Flowchart): This figure is well-designed but might benefit from an additional note that explains any specific reasons for excluding the 79 full-text articles.

Tables: Some tables (especially cost data tables) could benefit from clearer segmentation of inpatient vs outpatient settings and a summary statement at the bottom to highlight key findings.

Overall Evaluation:

This systematic review contributes valuable insights into the cost-effectiveness of penicillin allergy testing, but it needs some fine-tuning in structure, presentation, and clarity to fully convey its findings. With clearer transitions, better-structured tables, and a stronger conclusion, this manuscript would be a solid contribution to the literature on antibiotic stewardship and healthcare cost management.

Let me know if you'd like me to make specific revisions directly to the document!

Reviewer #2: The manuscript is a well-executed systematic review on an important healthcare topic. The conclusions drawn are appropriate given the available evidence; however, a greater emphasis could have been made on describing the limitations of studies reviewed, particularly in relation to data on health-related quality of life and long-term cost-effectiveness data. Revisions in the manuscript would encompass issues with respect to data availability, minor grammatical flaws, further description of study limitations, and future research directions.

Reviewer #3: I do enjoy reading the manuscript. However, this quality of this manuscript would have been improved by taking into consideration the following comments:

1. Please provide any specific reason to choose cost estimates in 2022 USD instead of 2023 USD (the timing when the running the full search)

2. I am bit confused with the different approaches in calculating median in Table 3. The median calculation was not consistent across the testing strategy. Please provide more explanation on this matter.

3. You are highly recommended to check the requirement of citation as some citations are not consistent i.e. some using abbreviation of journal name whilst others using the full journal name.

Reviewer #4: Overall, this article has demonstrated how a normal systematic review is conducted. The PRISMA technique is also employed for data extraction, and articles are evaluated using a relevant checklist. However, I believe that some aspects need to be changed (particularly in the method) so that the conclusions gained can be clearly established in accordance with the article's title and generalized.

The following are suggestions for improvements:

1. Abstract: the abstract's results do not correspond to the study's goal. This study examines costs, health-related quality of life, and the cost-effectiveness of penicillin allergy testing, however, the results include a variety of costs that I believe will make it difficult to draw meaningful economic conclusions. It would be preferable to include what cost categories are seen, what quality of life measurements are used (which do not appear in the results), and what effectiveness of costs might be improved.

2. Method: In the written research method, syntax is used extensively in the search (Supplementary file), but why is there no limit on the year of the article, why was the search carried out twice at a distant time (2020 to 2023), and is this search still relevant? It is advisable to choose the closest time range for data search (2023).

3. Data extraction cannot fully describe the articles obtained; it would be preferable if the characteristics of the articles were displayed (in the results section), such as authors, year of publication, location, and research design; additionally, it is necessary to display whether statistical measures such as OR, RR, or mean difference are used in the article.

4. You can include a more in-depth explanation of how the findings can be used in various circumstances, such as geographical or healthcare system disparities.

5. Data and presentation:

(a) Describe the sources of data heterogeneity. Explain, for example, if cost disparities are due to variability in test implementation or the setting of the health system.

(b) Quality of life: Although primary studies provide limited data, consider including a proxy or quality of life evaluation model. This can enhance the clinical and economic value of penicillin allergy testing

(c) Additional graphs, such as flow charts or histograms, can be used to depict cost trends, savings, and clinical outcomes of various test models.

6. Discussion:

(a) Discuss the cost-effectiveness of testing by non-allergy specialists (e.g., pharmacists or nurses) and how this approach could be broadly implemented, taking into account training, regulatory, and resource constraints.

(b) Provide a more thorough explanation for how de-labeling could help to reduce AMR, a major global concern.

(c) Please discuss the implications of these findings for policy and future study.

7. Article Language and Structure: It is preferable to use subheadings or bullet points in the discussion section to highlight key findings, such as cost-effectiveness or reduced length of stay.

8. This article could make a greater contribution to the scientific literature, both in terms of the quality of the analysis and its practical relevance. Authors are encouraged to highlight the strengths and limitations of the study more transparently while providing clearer evidence-based policy recommendations.

Reviewer #5: This manuscript is well-written. However, authors are advised to avoid using the term "effectiveness" in relation to the limitations of the effectiveness parameters found in the research articles. Additionally, the abstract section should include the total number of articles obtained from the search, specifying that 35 articles were ultimately reviewed.

Reviewer #6: This manuscript evaluates the cost-effectiveness of penicillin allergy testing, offering important insights into this critical area. Notably, the review highlights a gap in the existing literature: most studies fail to meet rigorous methodological standards and primarily emphasize direct healthcare costs, such as those related to antibiotic usage. Despite these limitations, the findings provide evidence supporting the role of penicillin allergy testing in improving the rational use of antibiotics, reducing healthcare costs, and enhancing patients' quality of life. Such contributions are critical for advancing antimicrobial stewardship and optimizing healthcare resource utilization.

Comments for Revision:

1. Line 113-114:The manuscript contains several typographical errors, including the incorrect use of colons. Please proofread the text carefully to correct these and any other typographical issues throughout the manuscript.

2. Line 170: How do the authors ensure that the publication quality assessment process is free from bias? Is the assessment carried out in an objective manner? Please include a detailed description of the risk of bias (RoB) assessment method in the manuscript. The results of this assessment should be included in the Supplementary Information section, and the data should be visualized within the manuscript for better comprehension.

3. Line 199-200: The sentence, “This yielded 35 unique articles reporting on 36 studies meeting the final inclusion criteria,” is unclear. According to Figure 1 (PRISMA Flow Diagram), it appears that 71 articles were included in this review. Please clarify this statement and ensure consistency between the text and Figure 1.

4. Line 211-212: Please move the descriptive summary characteristics of the studies, currently found in Tables S1 and S2, directly into the manuscript text. This will facilitate better understanding of the data and provide a more contextual basis for the discussion.

5. The emphasis of the findings of this study needs to be clarified in the discussion section. Besides, the presented conclusions undermine the results and do not adequately address the targeted problems. The author should revise the conclusions in both the abstract and conclusion sections. Additionally, the author must propose a strategy design for medical personnel to enhance the cost-effectiveness of antibiotic use for both PAL and non-allergic patients. This strategy should include practical solutions to address the mislabeling of penicillin allergies. These can be included after the discussion section and visualized as a chart to provide a straightforward, actionable resource for healthcare personnel.

Reviewer #7: This is a useful and interesting paper that is generally well presented.

Introduction

The introduction provides a strong foundation for the study, but it could benefit from improved contextualization of AMR, expanded examples of health risks, and an earlier emphasis on cost-related considerations. Refining the aim statement will further strengthen its alignment with the study's objectives and potential impact.

Methodology :

Line 171 : While the protocol deviation is mentioned, the rationale for choosing the published checklist over the CHEERS statement is not adequately explained. Providing more detail on why the checklist is more suitable for these studies would strengthen the justification. Add a sentence explaining why the published checklist was deemed more appropriate. Example: "We opted for a published checklist instead of the CHEERS statement because the latter includes numerous reporting elements that extend beyond the methodological requirements specific to economic evaluations in this area." or Briefly describe the key domains assessed by the checklist. Example: "The checklist focused on domains such as study design, data sources, cost estimation, outcome measurement, and sensitivity analysis, which align closely with the objectives of this review?"

The process for resolving conflicts by discussion is mentioned. A more detailed explanation would enhance transparency. Were conflicts solely resolved through discussion between the two reviewers, or was an impartial third party engaged to provide a final decision in cases where consensus could not be reached?

Data Presentation:

Tables or figures summarizing key cost metrics and study characteristics would enhance clarity and allow for easier comparison across studies.

Line 353 : The critique that "the majority met less than half the methodology quality appraisal items" is significant but underexplored. Specify which quality criteria were most commonly unmet (e.g., absence of control groups) and discuss how these deficiencies impact the reliability of the findings.

Line 407 : While the section mentions that de-labelling reduces hospital stay costs, it is unclear what the average magnitude of savings is and how it compares to the costs of testing. Quantify these effects wherever possible to strengthen the argument.

Line 446 : The call for studies with longer follow-up periods is valid but could be expanded. Suggest specific research designs or methodologies that might address the identified gaps, such as multicenter trials, cost-utility analyses, or studies incorporating patient-reported outcomes.

**Do you want your identity to be public for this peer review?** For information about this choice, including consent withdrawal, please see our Privacy Policy

Reviewer #1: **Yes: ** Dr Mahdy A A Osman

Reviewer #2: **Yes: ** Dr. Anant Kumar Patel

Reviewer #3: **Yes: ** Dr. Hesty Utami Ramadaniati

Reviewer #4: No

Reviewer #5: **Yes: ** Vitarani Dwi Ananda Ningrum

Reviewer #6: No

Reviewer #7: No

---

## [Author Response · Author response to Decision Letter 1]

29 Apr 2025

No. Comment Response Status

1 1. Please ensure that your manuscript meets PLOS ONE's style requirements, including those for file naming. 1. Please ensure that your manuscript meets PLOS ONE's style requirements, including those for file naming. The PLOS ONE style templates can be found at https://journals.plos.org/plosone/s/file?id=wjVg/PLOSOne_formatting_sample_main_body.pdf and https://journals.plos.org/plosone/s/file?id=ba62/PLOSOne_formatting_sample_title_authors_affiliations.pdf

We have reformatted the manuscript and files to meet the journal style requirements as indicated in the two links. Done

2 2. Please note that funding information should not appear in any section or other areas of your manuscript. We will only publish funding information present in the Funding Statement section of the online submission form. Please remove any funding-related text from the manuscript. We have deleted the funding information from the manuscript Done

3 3. Thank you for stating the following financial disclosure: “NIHR PGfAR RP-PG-1214-20007”.

Please state what role the funders took in the study. If the funders had no role, please state: "The funders had no role in study design, data collection and analysis, decision to publish, or preparation of the manuscript." If this statement is not correct you must amend it as needed. Please include this amended Role of Funder statement in your cover letter; we will change the online submission form on your behalf. We have incorporated this statement that the funder had no role in any aspect of the design and conduct of our study and publication in the Covering Letter of our resubmission. Done

4 4. In the online submission form, you indicated that “The detail data extracted from the studies included in the review are available from the authors upon request”.

All PLOS journals now require all data underlying the findings described in their manuscript to be freely available to other researchers, either 1. In a public repository, 2. Within the manuscript itself, or 3. Uploaded as supplementary information. We have presented the tables of extracted information in a new supplementary appendix (‘Supplementary S3 File, Table S4, Table S5, Table S6’). Done

5 5a. As required by our policy on Data Availability, please ensure your manuscript or supplementary information includes the following: A numbered table of all studies identified in the literature search, including those that were excluded from the analyses. For every excluded study, the table should list the reason(s) for exclusion. If any of the included studies are unpublished, include a link (URL) to the primary source or detailed information about how the content can be accessed. The table of all studies identified in the literature search for full text screening (Table S1), including those excluded from analysis and reasons for exclusion are already presented in Supplementary File 3.

Please note only published studies were included in the review. No change has been made.

6 5b. …A table of all data extracted from the primary research sources for the systematic review and/or meta-analysis. The table must include the following information for each study: Name of data extractors and date of data extraction Confirmation that the study was eligible to be included in the review. All data extracted from each study for the reported systematic review and/or meta-analysis that would be needed to replicate your analyses. If data or supporting information were obtained from another source (e.g. correspondence with the author of the original research article), please provide the source of data and dates on which the data/information were obtained by your research group. We have added this table as a supplementary appendices Table S4, Table S5, Table S6. Done

7 5c …. If applicable for your analysis, a table showing the completed risk of bias and quality/certainty assessments for each study or outcome. Please ensure this is provided for each domain or parameter assessed. This should be provided for each outcome. Risk of bias assessment results based on the ROBINS-I tool of non-randomised studies of interventions is now presented for each reviewed outcome as and where they appear in the Results. Done

8 5d…. An explanation of how missing data were handled. This information can be included in the main text, supplementary information, or relevant data repository. Please note that providing these underlying data is a requirement for publication in this journal, and if these data are not provided your manuscript might be rejected. Text has been added before the last paragraph of Methods:

‘The analysis was based on data as reported in the original publications, except when sufficient data were reported to calculate statistics for outcomes of interest, e.g. mean cost differences on a per patient basis. Only complete case analysis was conducted; i.e. no attempt was made to impute or otherwise adjust for missing data.’ Done

9 Key Strengths:

Demonstrates a standard systematic review using PRISMA.

Highlights the gap in literature, particularly in the methodological quality of primary studies.

Contributes to antimicrobial stewardship by emphasizing the role of penicillin allergy testing in rational antibiotic use, cost reduction, and quality of life improvement.

10 Areas for Improvement:

Abstract:

Results in the abstract are misaligned with the study's goal.

Clarify the cost categories, quality of life measurements, and effectiveness of costs examined.

The Results section of the abstract has been revised to read

“Thirty-sixfive studies met the inclusion criteria. Most studies analysed the effect of testing on the costs of antibiotic use among patients admitted to hospital with a PAL. Studies measured costs of testing (n=19); antibiotic medication use (n=23); adverse reactions with penicillin use (n=4), alternative antibiotic drugs (n=3); length of hospital stay (n=5); subsequent health care use episodes (n=4); and antibiotic medication use in subsequent care episodes (n=3). The median cost of skin testing plus oral challenge across sixfive primary costing studies was USD 246268 (range: 164153, 514480), which contrasts with the USD 42-258367-591 range of antibiotic cost savings during the initial hospital admission. The median reduction of costs of antibiotic therapy was 57% (range: 52-66; 3 cohort studies with concurrent controls) and costs of length of hospital stay of initial admission 35% (range: 23-47, two prospective studies)... Two studies presented evidence that penicillin allergy testing is cost-saving in an outpatient setting over 3.5-4.5 years. One model-based study reported that testing in inpatient settings is cost-saving. No reports on the effect of penicillin allergy testing on health-related quality of life were found and the two cost-effectiveness studies that accounted for this outcome employed the opinion of healthcare professional or an assumption of a common generic value for adverse reactions.” Done

11 Methodology:

Justify the lack of year restrictions and rationale for using a specific search timeline (2020-2023).

Added text in Methods:

We did not limit the year of our searches as we sought to maximise evidence capture on diversity in service models.

The time points of the update reflected our overall programme of research timelines of which this review was a part. We have not added any text on this but please let us know if you still think this is relevant.

Done

12 Provide an explanation for using 2022 USD instead of 2023 USD for cost estimates.

Include more details on the risk of bias (RoB) assessment and ensure results are visualized in the manuscript.

We have now updated our cost figures to reflect 2024 USD prices

Done

13 Explain the rationale for selecting a published checklist over the CHEERS statement and the key domains it addresses.

We have now added further assessments of RoB for observational outcomes (healthcare costs) using the ROBINS-I tool recommended by Cochrane Collaboration. This information has been added in Methods, Tables of results and main text in results, and Table S5 and Table S6.

We have added the following text in Methods:

“In a slight deviation from the protocol, we evaluated the methodological quality of the included economic evaluations using a published checklist designed for conducting systematic reviews (Evers et al. 2005) instead of the CHEERS statement (Husereau et al. 2013), which is a reporting checklist as opposed to a quality assessment tool”

Done

14 Clarify the conflict resolution process during data extraction and analysis.

We have added the underlined text in Methods:

“Study selection

Two independent reviewers initially screened the title and abstract of each identified record. The full texts of those deemed potentially relevant were then reviewed against the eligibility criteria to identify studies for inclusion. All screening was conducted in Rayyan (Ouzzani et al. 2016). Disagreements were resolved by discussion.

Data extraction

Study characteristics were extracted from each included study, including publication details, the type of study conducted, the population included in the study, the intervention and comparators. Any results reported on health outcomes and cost measures, total or incremental costs were also extracted. Data extraction was performed by one reviewer and independently by a second, with any discrepancies resolved by the opinion of a third reviewer.” Done

15 Data Presentation:

Display study characteristics (authors, year, location, design) in the results section.

This has now been included as Table 1 and Table 2

Done

16 Provide statistical measures (e.g., OR, RR, mean difference) where applicable.

Table 2 (now Table4 ) has had the label ‘Median’ revised to ‘Median of study means’. Table 3 (now Table 5) has had the label ‘Median’ revised to ‘Median of study mean differences’; same for Table 4 (now Table 6)

Done

16 Address heterogeneity sources, such as variations in testing implementation or healthcare systems.

Have added text in Results 2nd paragraph : “Observational studies were from the US (n=17), Australia (n=3), Spain (n=3), New Zealand, Canada, Denmark, Ireland and France. Model-based evaluations related to the US (n=6); Canada (n=2) and Europe (one evaluation that reported results for US and Europe).”

Done

17 Consider including proxy models for quality of life evaluation.

We considered but decided against doing this, simply because quality of life measures vary widely and are poorly reported in any literature making comparisons difficult.

No change made.

18 Enhance visual presentation with graphs like flow charts, histograms, or comparison tables.

We have added comparisons tables S4, tables S5 table S6.

Done

19 Discussion:

Expand on cost-effectiveness findings, including implications for non-allergy specialists conducting tests.

Address potential contributions of de-labeling to reducing antimicrobial resistance (AMR).

Discuss policy implications and propose actionable strategies for healthcare personnel.

Quantify savings from de-labeling compared to testing costs.

Suggest research designs to address gaps, such as cost-utility analyses and multicenter trials.

We have added the following text in the paragraph before Discussion:

“In addition to health-related quality of life, there is lack of evidence on long-term benefits of appropriate de-labelling to patients from avoiding delayed treatment of serious infections, e.g. sepsis or meningococcal meningitis. Further none of the studies identified in this review were designed to account for the population health benefits of PADL from reducing antimicrobial resistance. Costs associated with rare events leading to ICU care were almost absent form studies, reflecting the single study nature of most of them. Further, multicentre controlled studies are needed to generate evidence on these or adequate surrogates of these key outcomes, which would then serve to inform cost-effectiveness and cost-utility studies that account for outcomes relevant to patients as opposed to single provider institutions as in most of the received literature.

Our study suggests that more efficient penicillin allergy testing models that rely of trained non-allergy specialist are required in order for their adoption to appeal to service managers and payers. Such adoption would be easier and less costly for hospitals with robust electronic health record systems, telemedicine services and established networks with allergy testing centres. Whilst pharmacists may have the training most suitable to lead in this role of expanding access to testing services, they would cost more than suitably trained nurses or healthcare assistants.” Done.

20 Conclusions:

Strengthen conclusions by aligning them with study goals and presenting evidence-based recommendations.

Highlight practical strategies to address penicillin allergy mislabeling.

We have revised the Conclusions by inserting the underlined text:

“There is limited evidence that penicillin allergy testing results in antibiotic costs savings, as published studies are observational, often uncontrolled and poorly reported. Whilst there is evidence of reduced antibiotic consumption in the community, little is known about the associated cost impact of de-labelling. Penicillin allergy testing is unlikely to recoup its costs within the first tear after testing. There is emerging observational evidence to suggest significant cost savings to the health care system from reducing outpatient attendances and inpatient stays three to four years after testing. No evidence exists on the health-related quality of life impact of penicillin allergy testing. Future randomised controlled trials are required to address questions ofidentify optimal models penicillin allergy testing models, with sufficiently long follow-ups and power to detect meaningful impacts to patients and national health services. Non-allergy specialist delivery models may offer an affordable way to expand allergy testing service coverage beyond the limited capacity of allergy testing centres.” Done

21 Article Language and Structure:

Revise typographical errors and ensure consistent citation formatting.

We have corrected and marked in tracked changes throughout the document Done

22 Additional Comments:

Specify the median calculation approach in Table 3 and ensure consistency across strategies.

We have added the underlined text in the last paragraphs of Methods:

“Intervention effect on healthcare cost estimates from primary data reported by more than one independent study were converted to percentage change units and summarised in terms of minimum, median and maximum mean reported values across studies.” Done

23 Integrate supplementary data (Tables S1 and S2) into the manuscript for better context.

Include the total number of articles reviewed in the abstract and results. Have now included these tables in the main text as Tables 4 and 5.

Please note that the total number of articles reviewed was included in the first sentence of Results section of the abstract, and they were also included Results, after Fig 1 Study selection. Done

Reviewer #1:

24 The manuscript titled "The cost-effectiveness of penicillin allergy testing: a systematic review" is well-organized, but it requires some revisions to improve clarity, consistency, and presentation. Below are my detailed peer review comments:

Title and Abstract:

Title: The title reflects the content accurately, though it may benefit from a clearer focus on the main findings. For example, consider refining it to “Cost-effectiveness of Penicillin Allergy Testing: Evidence and Gaps from a Systematic Review.”

Have added the underlined text to the title: “The cost-effectiveness of penicillin allergy testing: evidence and gaps from a systematic review

Done

25 Abstract: The abstract is concise and informative. However, the conclusion seems cautious. You might consider

---

## [Decision Letter · Decision Letter 1]

8 Jul 2025

Dear Dr. Mujica-Mota,

Thank you for submitting your manuscript to PLOS ONE. After careful consideration, we feel that it has merit but does not fully meet PLOS ONE’s publication criteria as it currently stands. Therefore, we invite you to submit a revised version of the manuscript that addresses the points raised during the review process.

We look forward to receiving your revised manuscript.

Kind regards,

Muhammad Shahzad Aslam, Ph.D.,M.Phil., Pharm-D

Academic Editor

PLOS ONE

Journal Requirements:

Reviewers' comments:

Reviewer's Responses to Questions

**Comments to the Author**

Reviewer #5: All comments have been addressed

Reviewer #6: All comments have been addressed

2. Is the manuscript technically sound, and do the data support the conclusions?

Reviewer #5: Yes

Reviewer #6: Yes

3. Has the statistical analysis been performed appropriately and rigorously?

Reviewer #5: N/A

Reviewer #6: Yes

4. Have the authors made all data underlying the findings in their manuscript fully available?

Reviewer #5: Yes

Reviewer #6: Yes

5. Is the manuscript presented in an intelligible fashion and written in standard English?

Reviewer #5: Yes

Reviewer #6: Yes

Reviewer #5: 1. Introduction section: What are the gaps between this systematic review and other published systematic review articles on similar topics? What is the added value of this systematic review compared to others? (e.g. https://pubmed.ncbi.nlm.nih.gov/29355644/)

2. Figure 1: delete this phrase "Studies included in review (n =36 )" due to the article selection flowchart, after calculating the number of excluded articles, it should be 35, not 36. 35 or 36?  Authors need to firmly establish the inclusion and exclusion criteria for articles so that the results of the selected articles are clear and consistent.

3. Table 3. This table needs some tidying up especially due to some of the column headings need to be merged.

4.

Reviewer #6: Thank you for submitting the revised version of your manuscript.

Overall, I enjoyed reading the article, which effectively highlights the importance of a sustainable service delivery model to improve access to penicillin allergy testing and delabelling for patients who are not truly allergic. The current version provides an explanation and strengthens the earlier manuscript.

I simply have a minor comment regarding Figure 1. In its current form, the figure appears seems disorganized.

**Do you want your identity to be public for this peer review?** For information about this choice, including consent withdrawal, please see our Privacy Policy

Reviewer #5: **Yes: ** Prof. Dr. Vitarani Dwi Ananda Ningrum

Reviewer #6: **Yes: ** Honey Dzikri Marhaeny

---

## [Author Response · Author response to Decision Letter 2]

24 Sep 2025

“Clarify how this review differs from and adds value beyond existing systematic reviews on the topic. Explicitly state the gaps your work fills and its unique contributions.

Tidy up the layout by merging related column headings and improving alignment so that the table is clear and easy to interpret.”

We have addressed these questions through revisions in the Introduction and Discussion sections of the Manuscript and Revised Manuscript with Track Changes documents, as well as the revised Figure 1. Please find the detailed list of responses to all comments and associated revisions in the attached Responses to Reviewers file.

Please let us know any additional comments that you may have on these responses and revisions.

---

## [Decision Letter · Decision Letter 2]

8 Oct 2025

Dear Dr. Mujica-Mota,

Thank you for submitting your manuscript to PLOS ONE. After careful consideration, we feel that it has merit but does not fully meet PLOS ONE’s publication criteria as it currently stands. Therefore, we invite you to submit a revised version of the manuscript that addresses the points raised during the review process.

We look forward to receiving your revised manuscript.

Kind regards,

Muhammad Shahzad Aslam, Ph.D.,M.Phil., Pharm-D

Academic Editor

PLOS ONE

Journal Requirements:

**Additional Editor Comments:**

Your manuscript presents a valuable and well-structured systematic review on the cost-effectiveness of penicillin allergy testing. Both reviewers acknowledge the overall strength and relevance of the work, noting that most substantive issues have been addressed. However, several areas still require attention to improve clarity, consistency, and presentation. Specifically, the Methods section of the abstract should be rewritten for better structure and detail; redundant text between narrative and tables should be minimized; citations must be added where missing; and language and formatting inconsistencies should be corrected for smoother readability. Minor table clarifications—such as defining “PAL” and ensuring study dates are complete—should also be addressed. Once these refinements are made, the paper will be ready for acceptance.

Reviewers' comments:

Reviewer's Responses to Questions

**Comments to the Author**

Reviewer #8: (No Response)

Reviewer #9: (No Response)

2. Is the manuscript technically sound, and do the data support the conclusions?

Reviewer #8: Yes

Reviewer #9: Partly

3. Has the statistical analysis been performed appropriately and rigorously?

Reviewer #8: Yes

Reviewer #9: No

4. Have the authors made all data underlying the findings in their manuscript fully available?

Reviewer #8: Yes

Reviewer #9: Yes

5. Is the manuscript presented in an intelligible fashion and written in standard English?

Reviewer #8: Yes

Reviewer #9: No

Reviewer #8: The article is well written and of good standard. Previous reviewers' comments were well addressed. I only have a minor concern.

In Table 5 and 6, full meaning of PAL should be written under the tables.

Only 2 studied have dates of study in Table 1 and 2. Authors should check and clarify.

Reviewer #9: 1. The Methods section in the abstract is poorly written and lacks structure. It is recommended to revise this section to clearly describe the review design, inclusion criteria, and evaluation approach.

2. There is considerable repetition between the narrative and the tables, especially in the results section. For example, Table 1 and Table 2 already list the countries of included studies, yet the same information is repeated in the “Identified records and included studies” section. It is recommended to avoid duplicating content already presented in tables.

3. In the results section, the study quality assessment, percentages are reported without accompanying sample sizes. Please revise.

4. Several studies are mentioned in the introduction, results and discussion sections without citation. In the Introduction, “Several recent studies have reported safe and effective testing…”. On page 83, the sentence “Only a couple of studies, both based on decision tree models, reported results in terms of quality-adjusted life years (QALYs)” it also lacks references.

5. The Discussion section is generally well written and provides useful insights. However, the" Limitations section" requires thorough revision. These are specific points to consider:

On page 88, the authors list “only including studies that reported costs” as a limitation, which is misleading. This is the core objective of the review and aligns with the article’s title.

On page 88, the authors begin by listing limitations of the submitted review, then abruptly shift to critique the included studies — for example: “In addition to health-related quality of life, there is lack of evidence on long-term benefits of appropriate de-labelling to patients from avoiding delayed treatment of serious infections” — before returning to limitations of the review itself. Please revise.

The manuscript requires thorough language revision to improve clarity and flow. Several sections contain inconsistent sentence structure. Example - Abstract: “We conducted a systematic review of published economic studies of penicillin allergy testing.” Repetitive phrasing- “published economic studies” and “systematic review” already imply this. The Results section: “This yielded 35 unique articles reporting on 36 studies…” is confusing and needs clarification.

**Do you want your identity to be public for this peer review?** For information about this choice, including consent withdrawal, please see our Privacy Policy

Reviewer #8: No

Reviewer #9: No

---

## [Author Response · Author response to Decision Letter 3]

22 Oct 2025

Thank you for your comments. We have made revisions to the manuscript in response to the comments made from the two reviewers, as detailed in the Response to Reviewers document attached and reflected in the Manuscript and Manuscript with Tracked changes.

Please let us know if you have any additional questions.

RMM

---

## [Decision Letter · Decision Letter 3]

5 Nov 2025

The cost-effectiveness of penicillin allergy testing: evidence and gaps from a systematic review

PONE-D-24-34463R3

Dear Dr. Mujica-Mota,

We’re pleased to inform you that your manuscript has been judged scientifically suitable for publication and will be formally accepted for publication once it meets all outstanding technical requirements.

Kind regards,

Muhammad Shahzad Aslam, Ph.D.,M.Phil., Pharm-D

Academic Editor

PLOS ONE

Additional Editor Comments (optional):

Reviewers' comments:

Reviewer's Responses to Questions

**Comments to the Author**

Reviewer #8: All comments have been addressed

2. Is the manuscript technically sound, and do the data support the conclusions?

Reviewer #8: Yes

3. Has the statistical analysis been performed appropriately and rigorously?

Reviewer #8: Yes

4. Have the authors made all data underlying the findings in their manuscript fully available?

Reviewer #8: Yes

5. Is the manuscript presented in an intelligible fashion and written in standard English?

Reviewer #8: Yes

Reviewer #8: (No Response)

**Do you want your identity to be public for this peer review?** For information about this choice, including consent withdrawal, please see our Privacy Policy

Reviewer #8: No

---

## [Editor Report · Acceptance letter]

PONE-D-24-34463R3

PLOS ONE

Dear Dr. Mujica-Mota,

I'm pleased to inform you that your manuscript has been deemed suitable for publication in PLOS ONE. Congratulations! Your manuscript is now being handed over to our production team.

Kind regards,

on behalf of

Dr. Muhammad Shahzad Aslam

Academic Editor

PLOS ONE